# 1D Quantum Simulations of Electron Rescattering with Metallic Nanoblades

**Joshua Mann \*** , **Gerard Lawler** and **James Rosenzweig**

Department of Physics and Astronomy, University of California, Los Angeles, CA 90024, USA;
gelawler@physics.ucla.edu (G.L.); rosen@physics.ucla.edu (J.R.)
*   Correspondence: jomann@physics.ucla.edu

**Abstract:** Electron rescattering has been well studied and simulated for cases with ponderomotive energies of the quasi-free electrons, derived from laser–gas and laser–surface interactions, lower than 50 eV. However, with advents in longer wavelengths and laser field enhancement metallic surfaces, previous simulations no longer suffice to describe more recent strong field and high yield experiments. We present a brief introduction to and some of the theoretical and empirical background of electron rescattering emissions from a metal. We set upon using the Jellium potential with a shielded atomic surface potential to model the metal. We then explore how the electron energy spectra are obtained in the quantum simulation, which is performed using a custom computationally intensive time-dependent Schrödinger equation solver via the Crank–Nicolson method. Finally, we discuss the results of the simulation and examine the effects of the incident laser's wavelength, peak electric field strength, and field penetration on electron spectra and yields. Future simulations will investigate a more accurate density functional theory metallic model with a system of several non-interacting electrons. Eventually, we will move to a full time-dependent density functional theory approach.

**Keywords:** rescattering; virtual detector; strong field; enhancement factor; TDSE; FDTD; photocathode

## 1. Introduction

Electron rescattering from a metal is the process in which an intense ultrafast laser pulse incident on a metal frees electrons through a three step process as is described here. An electron begins bound to a metal, and an external laser field is applied with polarization normal to the metal. In order of required field intensity, the electron may multi-photon ionize [1], above-threshold ionize (ATI) [2], or quantum tunnel (strong-field ionize) [3] out of the metal. It then propagates in the field and either continues away from the system (direct electrons) or returns to the metal to be potentially scattered and re-emitted (rescattered electrons). This photoionization process is different from the standard photoelectric effect in that, with sufficient fields, quantum tunneling effects overpower the dimmer multiphoton and ATI contributions. These processes have been studied extensively in the atomic, or gas, regime ([4,5] for instance), and increasing interest has begun to surround metallic emitters. In the case of bulk metal emitters, experiments and numerical studies have been performed with metallic nanotips in order to achieve high enhancement fields [6]. However, these nano-scale tips are prone to damage and are limited by their point-sized emission surface. The UCLA extreme high brightness photocathode experiment aims to circumvent this issue by utilizing a blade nanostructure as opposed to a nanotip. The nanoblade structure provides superior damage thresholds, a 1D emission surface (significantly larger emission area than a single point, 0-D), and trivial scalability without sacrificing laser enhancement properties. These benefits combined permit high brightness electron beam production from cold, durable photocathodes manufactured using standard etching and deposition processes.



Previously, simulations have been performed with incident fields upwards of 50 GV/m for gas-based high harmonic generation (HHG) [7] and with time-dependent density functional theory (TD-DFT) for the metallic tip rescatter process [6]. However, works analyzing the light-metal interaction keep to modest peak fields and do not include important physical factors including field dropoff [8,9] and field penetration into the metal. Works that do include field dropoff only reach a peak enhanced field intensity of $5 \times 10^{12}$ W cm$^{-2} \approx 6.1$ GV/m [10], which is a considerable field considering the fragility of nanotips. In this paper, we introduce the various methods involved with the quantum simulation, including our method for measuring the electron spectrum and the electrostatic/dynamic potential models used. We analyze the results of the simulation and investigate the effects on spectra and yields from changes in peak field amplitude, wavelength, field penetration, and ground state energy. Various alterations to the model and approach are suggested as future work.

## 2. Methods

Electronic simulations are performed using the 1D time-dependent Schrödinger equation (TDSE) via the Crank-Nicholson method. The system is depicted in Figure 1, where a cross section of the blade is shown and the simulation space is represented as the dashed line. The ground state of the system is determined using imaginary time propagation [11]. Absorptive boundary conditions are included by applying a spatially varying complex multiplier to the kinetic energy operator in the Hamiltonian, i.e.,

$$\hat{H} = \tilde{\xi}(x)\hat{T} + \hat{V}(x). \tag{1}$$

This is effectively an added local imaginary time propagation near the simulation boundaries, with decay rate increasing linear with the kinetic energy of the incident electron. The complex function $\tilde{\xi}(x)$ takes a value of 1 within the main simulation and smoothly approaches to $1 - ci$ for some positive $c$ using a polynomial smooth function near the simulation boundary. The kinetic and potential energy operators are otherwise unchanged from the standard approach.

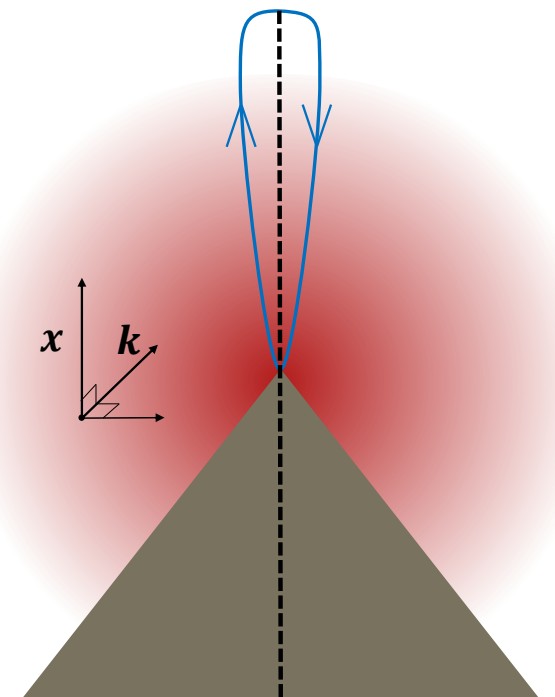

**Figure 1.** Cross section of the blade system. The laser's wavenumber is in direction **k** (into the page) and the simulation grid is shown as the dashed line, in the **x**-direction. Rescattering electrons follow a trajectory akin to that shown in blue (although with far less lateral motion if emitted from the blade edge).

## 2.1. Electrostatic Potentials

The electrostatic potentials used to model the metal's properties are the Jellium potential and a planar averaged shielded atomic potential. The Jellium potential with accurate bulk properties [12] and image charge potential [13], using atomic units, is given by [14]

$$
V_{jellium}(x) = \begin{cases} -\dfrac{V_0}{A\exp(Bx)+1} & x < x_{im}, \\ -k\dfrac{1-\exp(-bx)}{4x} & x \geq x_{im} \end{cases} \tag{2}
$$

where $V_0 = E_f + W$ ($E_f$ is the Fermi energy and $W$ is the work function), $x_{im} = -0.2r_s + 1.25$ is the image plane position, i.e., the distance between the first atomic layer and what is considered the edge of the metal. $r_s$ is the Wigner–Seitz radius (a sphere of this radius inside the metal should contain on average one electron), determined by relating two formulae for the bulk number density $n_{bulk} = \frac{1}{3\pi^2}(2E_f)^2 = \frac{3}{4\pi r_s^3}$. $b = k_f$ is the Fermi wave number calculated by $E_f = k_f^2/2$. $A = 4V_0/b - 1$ and $B = V_0/(4V_0/b - 1)$ affect the surface properties of the Jellium model. All of these quantities are standard in solid state physics, except for $A$, $B$, and $x_{im}$, which were determined by fitting to DFT results. $k$ is chosen to ensure continuity. $E_f$ and $W$ were chosen to be 9.2 eV and 6.2 eV, respectively, for standard simulations. This potential includes both the inner sigmoid-like potential ($x < x_{im}$), representing the metal's bulk and surface, and an external image charge potential ($x \geq x_{im}$).

The planar averaged shielded atomic potential is given by, in SI units,

$$
V_s = -\frac{Ze^2}{2\kappa\epsilon_0 d^2}e^{-\kappa|x|}, \tag{3}
$$

where $Z$ is ideally the number of protons in the metal, but as an effective potential it was chosen to be 1.74. $d$ is the mean spacing between the atoms in a square grid, with a value of 2.5 Å. $\kappa$ is the shielding parameter, chosen in this case to be 1 Å$^{-1}$. This atomic potential is centered at $x = 0$. The atomic potential and the Jellium potential are combined to produce the electrostatic potential used throughout all simulations, with some universal scaling when performing preliminary total yield calculations.

## 2.2. Simple Light Field Model

The electric field due to the incident light pulse is taken to be

$$
E(x,t) = E_{enh}(x)e^{-2\ln 2\frac{(t-t_{max})^2}{\tau^2}}\exp(i\omega_0(t-t_{max}) + i\phi_{ce}), \tag{4}
$$

where $\tau$ is the full width at half maximum (FWHM)-power of the pulse, $t_{max}$ is the time in the simulation of peak pulse, $\omega_0$ is the central frequency and $\phi_{ce}$ is the phase, chosen to be $3\pi/2$ in this paper, or sine-like where the electron is pulled out before being pushed back in at the peak of the pulse. $E_{enh}(x)$ is the (complex) function representing the electric field, including any enhancement effects from the blade structure. For simulations without field penetration, it takes the form outside of the metal (beyond $x_{im}$, this assumes a perfect conductor)

$$
E_{enh}(x) = \frac{E_{max}}{\beta_{max}}\left[\frac{R}{x - x_{im} + R}, (\beta_{max} - 1) + 1\right] \tag{5}
$$

where $E_{max}$ is the peak enhanced field (not the incident laser field), $R$ is the radius of the blade (20 nm), and $\beta_{max}$ is the enhancement factor. The exact value of $\beta_{max}$ (when keeping the enhanced field constant) does not affect the simulation much as long as it is significantly above 1. The existence of this enhancement factor and its decay from the metal induces a ponderomotive force away from the metal, aiding in emission. We choose an enhancement factor of 12 for these simpler simulations. The roughly $1/r$ dependence on field strength arises from the near-cylindrical shape of the blade

edge. The angle of the blade induces other fractional powers of $1/r$; however, $1/r$ is the dominant contributor. Simulations including field penetration directly use the complex fields calculated using Lumerical's FDTD Solutions (8.16.884, Lumerical, Vancouver, BC, Canada). These fields predict an enhancement factor of about 4; however, we believe experiments will have higher enhancement factors as surface abnormalities lead to hot spots [15].

The static Jellium and surface atomic potential with possible ranges including the external fields are shown in Figure 2. The ground state wave function is localized to the atomic potential, giving the atomic scattering potential extra utility as an anchor for the surface electron.

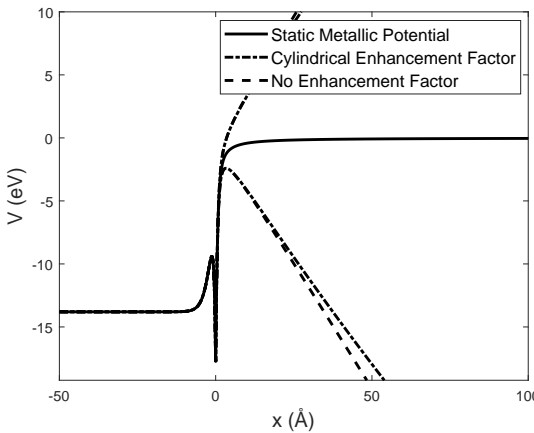

**Figure 2.** Electrostatic potential, $V_{jellium}(x) + V_s(x)$, with and without external fields applied. The ground state electron is largely bounded to the surface atomic potential.

### 2.3. FDTD-Based Light Field Model

Our finite-difference time-domain (FDTD) simulations of an 800 nm laser pulse incident on our nanoblade structure are performed using Lumerical. The calculations use a perfectly sharp triangular prism of silicon (with total internal edge angle of 54.7 degrees) with a 5 nm layer of titanium, followed by a 35 nm layer of gold resulting in an edge radius of 40 nm. The peak field normal to the blade surface (along the simulation space depicted in Figure 1), and its complex phase is shown in Figure 3. This result is what is used when performing simulations including field penetration. While the FDTD simulations include all layers as is pertinent for FDTD's accuracy, the TDSE simulations do not go deep enough into the metal to include these features.

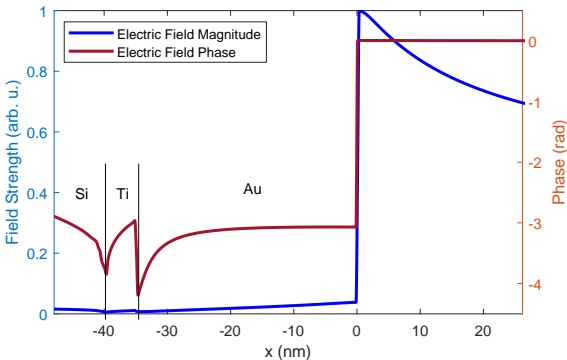

**Figure 3.** FDTD simulation of our Si-Ti-Au nanoblade. The peak field magnitude (blue) is mostly constant within the metal and follows a near $1/r$ profile outside the metal. The phase (red) includes a $\pi$ phase shift into the metal and some interesting behavior following. Our TDSE (time-dependent Schrödinger equation) simulations do not go far enough into the metal to include these effects as of yet. The first atomic layer is placed at $x = 0$ and so the surface of the metal is actually at $x = x_{im}$ in this plot.

### 2.4. The Virtual Detector Method for Measuring Emitted Electron Spectra

The virtual detector (VD) method [16,17] is a useful tool for measuring emission properties of a system. We place a virtual detector that measures both the wave function and the probability current near the simulation boundary and before the absorptive region. Thus, we obtain the wave function $\psi(r_{VD}, t)$ and the probability current $j(r_{VD}, t)$ at the VD position $r_{VD}$. These measurement processes (among others) may be performed in parallel with the TDSE solver. Our goal is to obtain the energy spectrum of emitted electrons. We begin by taking a Fourier transform (via a Tukey windowed fast Fourier transform) in time of our wave function, producing $\psi(r_{VD}, \omega)$. To get the number of electrons emitted, we need the probability flux of this wave function at each energy. To do this, we take the modulus squared of this function (to get the density of electrons near the edge at each energy), multiply by the velocity (to get the flux of electrons at each energy) and multiply by an undetermined factor to ensure normalization with the true emission. In summary,

$$|\psi(E)|^2 = N(E) = \eta\sqrt{E}\left|\int_{-\infty}^{\infty}\psi(r_{VD}, t)e^{\frac{-iEt}{\hbar}}dt\right|^2,\tag{6}$$

where we changed variables via $E = \hbar\omega$, and $\eta$ is a constant that can be determined by ensuring that the number of electrons emitted is consistent between the integrated probability current and the integrated spectrum, thus leading us to

$$N(E) = \frac{\int_{-\infty}^{\infty}j(r_{VD}, t)dt}{\int_{0}^{\infty}\sqrt{E}\,|\mathcal{F}_{t\to E}\psi(r_{VD}, t)|^2\,dE}\sqrt{E}\,|\mathcal{F}_{t\to E}\psi(r_{VD}, t)|^2,\tag{7}$$

where $\mathcal{F}_{t\to E}$ denotes the Fourier transform from time to energy. Although the absorptive boundaries should be sufficient to prevent reflections from the simulation edge, one may take only positive (or negative) frequencies from the FFT to ensure that only electron emissions and not boundary reflections are being included. Additionally, if the potential at the VD changes in time, one may divide the time-dependent wave function by the integrated complex phase (of magnitude 1) induced by the potential before performing this calculation, thus removing the contribution from the potential. The normalization via integrating the probability current fails when there is a significant probability flux in both directions (one could obtain zero electrons by this method if there were equal currents going in both directions, for instance), and in such a case one would not be able to normalize in this manner. While the factor $\eta$ can be calculated exactly for an analytic Fourier transform, the necessary application of a window to perform the numerical FFT unphysically alters the yield and therefore normalization via probability current is more robust in our case.

This method is compared with the window operator (WO) method [18] in Figure 4. The WO method determines the energy spectrum of the wave function at the end of the simulation. The VD method proves to be a good alternative to the WO method for measuring high energy electrons. The VD method seems to have a lower noise level while requiring very little post processing computation (VD requires an FFT while WO requires several tridiagonal matrix equation solutions). These two methods may hypothetically be combined to get good low energy and high energy results while keeping simulations spatially and temporally small in the future.

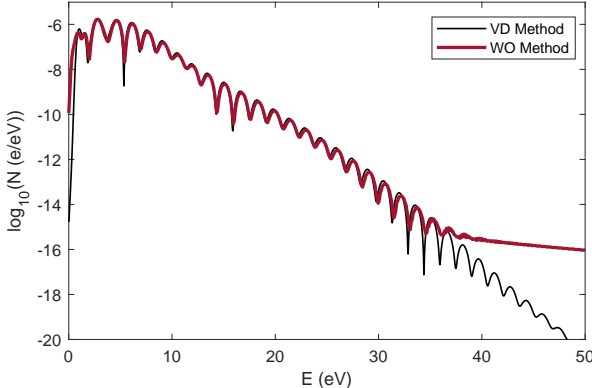

**Figure 4.** Window operator method (red, bold) and virtual detector method (black) compared for a gas electron rescattering test simulation with $\phi_{ce} = 0$. The WO (window operator) method ideally performs best in the low energies, where electrons have not had time to leave the simulation. However, the noise level (seen on the right as a flattening out) is fairly high and the WO method involves computationally intense post processing. The VD (virtual detector) method may not have measured all of the low energy electrons as they did not make it to the boundary by the end of the simulation. Regardless, the two spectra overlap closely and the VD method also seems to have a lower noise level, all while requiring just a single FFT (fast Fourier transform) in post processing.

## 3. Results

Simulations were performed using the Jellium with surface atom electrostatic potential with both the simple and penetrating fields. Figure 5 shows the probability density as a function of time for a peak field of 20 GV/m at 800 nm with no field penetration. The electron is shown to begin in its ground state. As the Gaussian laser pulse progresses, more and more of the electron wave function is able to be excited and/or tunnel into the field. Once in the field, it may propagate and either return to be scattered or simply leave the simulation directly.

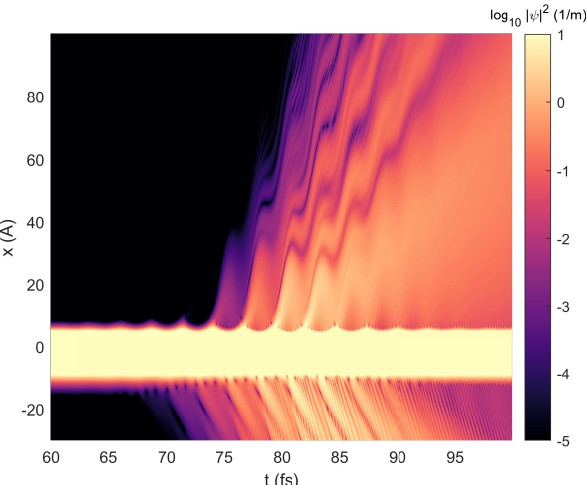

**Figure 5.** A focused view of the electron probability density in space over time. The electron begins at its ground state in the metal towards the left. As the external field is applied, the electron may be excited and/or tunnel out of the metal. Once outside the metal, it may propagate and either return to the metal or escape permanently. Various energy bands may be seen just from this image from the different slopes of electron probability density bunches. This simulation was performed using a 20 GV/m peak enhanced field at 800 nm. Note that the snippet shown here is only a piece of the full simulation—neither spatial nor temporal boundaries are included.

We compared spectra from our simulations to data from experiments to verify results [15]. We found that, likely due to the improper energetics of the simulation, we had to not only scale the data (as is generally expected), but we also had to exponentiate the data. By exponentiating the data, we account for the nonlinear decrease in electron yields as a function of ground state energy. Additionally, the surface layer in the experiment was made of tungsten, so the simulated field penetration (with a gold surface) is not exactly ideal. However, the existence of field penetration is important enough to see preliminary results, especially when considering the low ground state energy. In Figure 6, we compare experimental data to this scaled simulation data. While the direct electron (low energy) spectra do not line up well, the plateaus (rescattered electrons, high energy) do. In fact, this is a useful method of checking the peak laser intensity and, therefore, the enhancement factor of the experiment. By matching the experiment data to the simulation that best fits the plateau, we ultimately find the peak field required to produce such spectra. While the exact numbers coming out of the simulation may not be of much use in lieu of this comparison, the energy axis and the qualitative plot shapes do seem to match well and can therefore provide insight on how to continue.

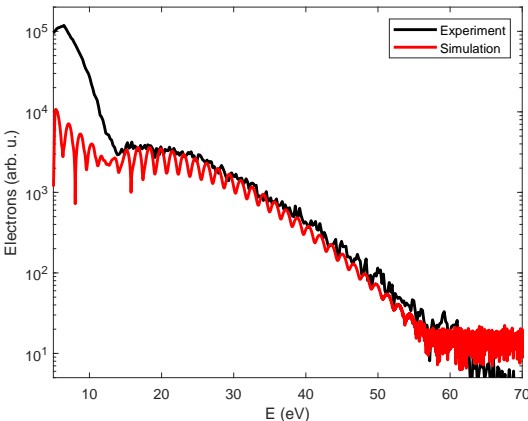

**Figure 6.** A comparison of experimental [15] and simulation data. While both scaling by a constant and exponentiating the data, we arrive at plots that largely agree in the plateau regime. Note that the energy axis is only translated, not scaled, to account for the bias voltage in the experiment.

We performed several simulations with varying parameters. As seen in Figure 7, with higher incident fields come higher energy and intensity electron emissions. In the low to mid field regime (below 20 GV/m), we see a standard exponential decay transitioning to a plateau, with a final dropoff. This low field region is focused on to the right in Figure 7. We begin to see the makings of a plateau emerging form the direct electron spectrum with fields as low as 8 GV/m, which corresponds to a reasonable peak enhanced intensity of $8.5 \times 10^{12}$ W/cm$^2$. However, as we reach higher electric fields (above 20 GV/m or $5.3 \times 10^{13}$ W/cm$^2$), we begin to see interesting hump structures, which arise from expected resonances due to channel-closing when we reach high peak intensities, as seen in atomic ATI [19].

A comparison of how the ground state energy and inclusion of field penetration alter the electron spectrum is shown in Figure 8. While field penetration does not have much effect on the spectral shape, as the process is largely the same, altering the potential to fix the ground state does. By scaling the potential to bring down the ground state energy from –19.8 eV to –4.7 eV, we effectively make both multiphoton absorption and quantum tunneling more probable, thus providing higher yields. The shape is also slightly altered in that the direct electrons overpower the rescattered electrons for higher energies (the transition is at 18 eV instead of about 10 eV). Finally, the $\hbar\omega$ peaks are more pronounced in the modified potential's dropoff curve.

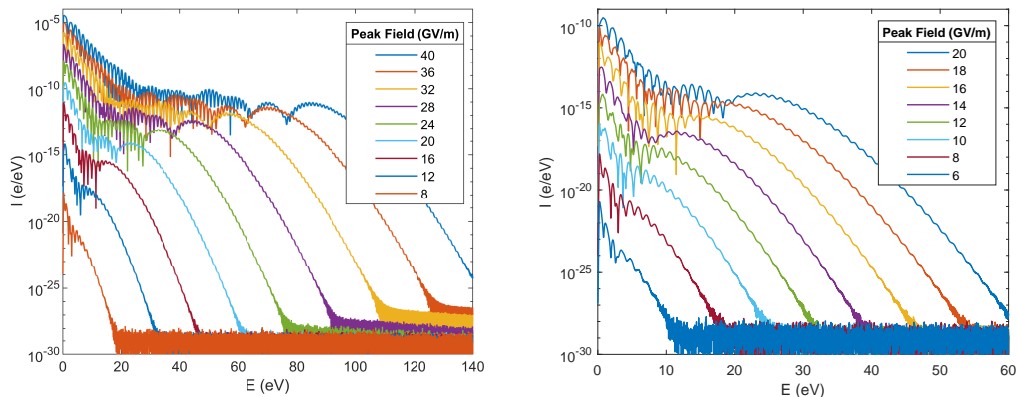

**Figure 7.** Comparison of spectra among all peak fields simulated (**Left**) and among mid to low fields (**Right**). These simulations use the standard potential with 800 nm incident laser pulse, no field penetration.

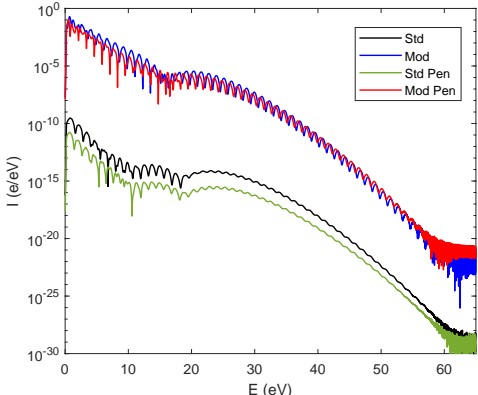

**Figure 8.** Comparison of standard spectrum (Std) against the modified potential (Mod) (scaled to set ground state to 4.7 eV), penetrating field (Std Pen), and the penetrating field with modified potential (Mod Pen).

We performed simulations for various electric fields and wavelengths and produced variable spectra shown in Figure 9. When keeping the wavelength constant and varying the peak field, we observe a quadratic increase of the peak electron energy (linear with enhanced intensity). This corresponds well with the semi-classical electron cutoff energy prediction of about $10U_p \propto \lambda^2 E_{max}^2$. When varying the wavelength and keeping peak field constant, we observe a slight deviation from the semi-classical cutoff. The variations in peak-to-peak wavelength and peak field induced by the 8 fs FWHM-power Gaussian envelope are included in the ponderomotive energy calculations, and are therefore unrelated to the deviation observed. High wavelength experiments (1800 nm or longer) and high field (20 GV/m or higher) will be needed to verify this phenomenon.

The electron yields from our simulations are shown in Figure 10. From gas-based ATI, we expect a nonlinear power law in our yield curves (for instance, a nonlinearity of about 2.8 in [20]). Since the yield strongly depends on the energy of the electron within the metal, and since we are stuck with using the ground state (we are not using a slab geometry and we have absorptive boundary conditions), we simply scaled the electrostatic potential such that the ground state is at the modest value of −4.7 eV, around the work function of most metals. In the electron rescatter process, we expect to observe three regimes, in order of dominance at lower to higher incident intensities: multiphoton absorption, quantum tunneling, and space-charge limited. During multiphoton absorption, we expect to see a power law according to the number of photons needed for absorption. With 800 nm light and the given ground state energy, we expect to require either three or four photons (in terms of energy, just slightly over three photons are required). Following this is the tunneling regime which begins to take hold

at Keldysh parameter $\gamma < 1$, which happens around $4 \times 10^{13}$ W/cm$^2$. This region matches best with the second transition from nonlinearity 2–2.5 to about 1 in Figure 10 (with field penetration). The last regime, being space charge, should not be represented here as the electron-electron interactions were not included in this model. Transitions may also occur as fewer-photon absorption occurs and these slightly excited electrons may more easily tunnel out of the field. Once an electron has absorbed one photon, its effective Keldysh parameter is now 1 at $2.6 \times 10^{13}$ W/cm$^2$, which is closer to the first transition in the same yield plot. While one cannot attribute the shape of these plots to these regimes without a doubt, this serves as a light interpretation.

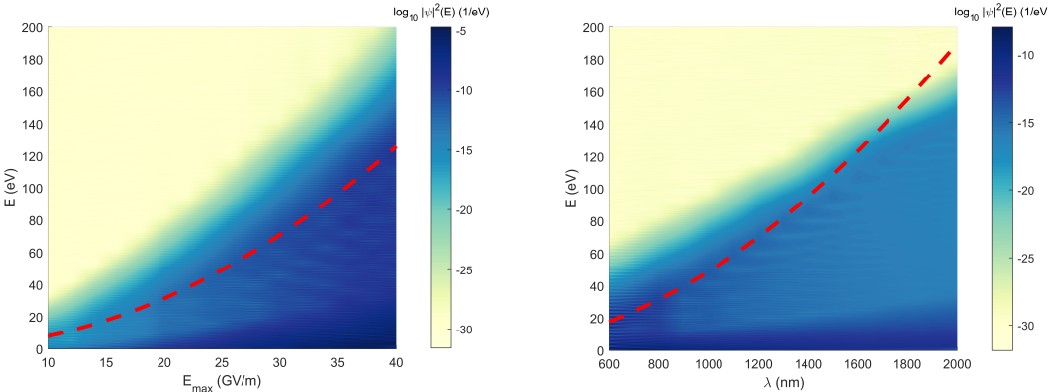

**Figure 9.** Electron spectra for varying peak field and wavelength. The semi-classical cutoff is shown as a dashed red line. Variable peak field at 800 nm (**Left**) shows a good trend of following the classical $10U_p \propto E_{max}^2$ cutoff for electron rescatter spectra. Spectra at variable wavelength with peak field of 20 GV/m (**Right**) also indicates a good general trend to the classical cutoff of $10U_p \propto \lambda^2$. This spectral map begins to deviate from the prediction beyond 1600 nm.

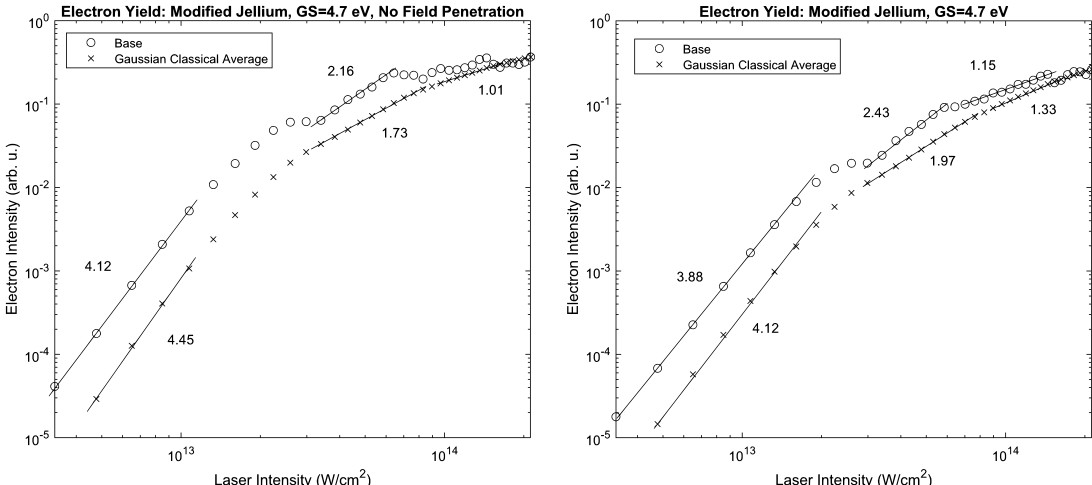

**Figure 10.** Simulated electron yield using a modified potential with incident laser wavelength of 800 nm. The electrostatic potential (excluding the external field) was scaled such that the ground state was at −4.7 eV, a much more reasonable value for a metal's work function. The yields with (**Right**) and without (**Left**) field penetration are shown. The individual simulated yields are shown as o's and the integrated yields (averaged over a 2D Gaussian intensity profile) are shown as x's. Experiments have shown power laws of about 2.5, tapering to 1 [15]. While we do not see those exact values here, we do see regime transitions. Note that the reported laser intensity is the enhanced intensity as opposed to incident.

## 4. Conclusions

We performed 1D TDSE simulations of an electron bound to a metal undergoing electron rescattering. We introduced some standard electrostatic potentials and light field models to best match the experiments. We also used some FDTD-derived fields to look for field penetration effects. To obtain electron spectra, we summarized a modification of virtual detector theory for simple emitted electron spectrum calculations. This method matches the window operator method almost exactly, and has a lower noise level, while requiring less computation time in post. We analyzed the simulation results by taking a qualitative glance at the probability distribution and spectra, noting how the spectra change with light wavelength, enhanced peak field, field penetration, and ground state modification. The electron yield plots show that there are several regimes in the simulation being modeled. However, it is uncertain as to what the power laws determined exactly represent and what their specific values entail. Future simulations will look into using DFT-derived slab effective potentials. One of the largest pitfalls of our model is the trade-off between the accuracy of the Jellium depth and the ground state energy (which is ideally at the Fermi level). Near-future simulations will include several non-interacting electrons with initial states from the bottom of the slab's potential up to the Fermi level. Additional analyses of high harmonic generation (HHG) will be performed with this model and the slab model. Far-future simulations will likely use TD-DFT in two or three dimensions.

**Author Contributions:** Investigation, J.R.; Software, J.M. and G.L.; Supervision, J.R.; Writing-original draft, J.M.

**Funding:** This research was funded by the Center for Bright Beams, National Science Foundation Grant No. PHY-1549132.

**Acknowledgments:** We would like to thank Ryan Roussel, Yusuke Sakai, and Oliver Williams for managing the experiment side of our HHG/electron rescattering operation at UCLA, as well as the undergrads who contributed to it (River Robles, Kunal Sanwalka, Victor Yu, and Cathy Zhuang). We would also like to thank Timo Paschen and Peter Hommelhoff at Friedrich-Alexander-Universität Erlangen-Nürnberg for accompanying us on the road to HHG/electron rescattering. Finally, we would like to thank the collaborators in the Center for Bright Beams, namely Tomas Arias at Cornell and Siddharth Karkare at ASU, who were helpful in determining how to continue development.

**Conflicts of Interest:** The authors declare no conflict of interest.

## Abbreviations

The following abbreviations are used in this manuscript:

| | |
|---|---|
| ATI | Above Threshold Ionization |
| DFT | Density Functional Theory |
| FDTD | Finite-Difference Time-Domain |
| FFT | Fast Fourier Transform |
| FWHM | Full-Width at Half-Max |
| HHG | High Harmonic Generation |
| TD-DFT | Time-Dependent Density Functional Theory |
| TDSE | Time-Dependent Schrödinger Equation |
| UCLA | University of California at Los Angeles |
| VD | Virtual Detector |
| WO | Window Operator |

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
