# Peer review of "1D Quantum Simulations of Electron Rescattering with Metallic Nanoblades"

_instruments, doi:10.3390/instruments3040059_

Round 1
Reviewer 1 Report
The authors present the results of one-dimensional simulations of above-threshold ionization (ATI) of an electron bound to a model potential at the surface of a metal tip (a blade in 1d) and driven by an infrared laser pulse. This is an interesting and timely topic, and the results may be worth being communicated. However, in its current form the paper is far from suitable for publication.
My list of complaints is long (and still incomplete as laid down below).
ATI in the gas phase arguably is the best-investigated strong-field phenomenon in atomic physics. Of course, it provides the background to the current paper. The results for metals largely derive their interest by comparison with the atomic case. Yet, the paper does not contain a single reference to atomic ATI.
In general, the list of references is completely inadequate. Just to begin with: Reference 1 is an unpublished 5-years' old Ph. D. thesis. Presumably, the important results of the thesis have been published in the mean time. Indeed, there is an extensive Topical Review (JPB 51, 172001 (2018)) with the thesis author on the list of authors. So why not cite this or other published papers by Wachter? Reference 2 is on HHG rather than ATI, it is focused on attosecond phenomena, which are irrelevant to the current paper and, moreover, it is hard to access, and so on ...
There is also a lot of previous literature on ATI in metals. How does the present work fit in? What are the differences if any? Which problems unsolved as yet are now resolved?
The discussion of the model potential is unsatisfactory. The physical meaning of the various parameters should be discussed and not just their numerical values be given. So, what is the meaning of x_im, what are W, and r_s? Presumably, the potential plotted in Fig. 1 is V_jellium + V_s. If so, why not write it down, and what are the parameters underlying Fig. 1?
The field (4) is like the incident field, except for a different space-dependent amplitude. What is the relation between the amplitudes of the incident field and the enhanced field at the tip?
The authors compare their simulation results with "experiments", but do not disclose what these experiments are.
Figure 7 exhibits "interesting hump structures", which the authors trace to "emissions earlier in the pulse" or to "perturbations from the semiclassical 3-step model approach". Actually, they are probably the same interference structures that are well known and have been much discussed in atomic ATI and can be traced to interference of different quantum orbits.
What are the dashed red lines in Fig. 9 and what is the reason of the discrepancy between the atomic 10 U_p cutoff and the cutoffs of the metallic ATI spectra?
The total yields of Fig. 10 should be compared with the atomic case. There are many such calculations available in the published literature.
The acknowledgment was written by the first author who expresses his appreciation of support by the other two. This is not the purpose of an acknowledgment.
In summary, the paper requires very extensive revision. It cannot be the job of the referee to provide the authors with a list of relevant previous works and references. I withhold a recommendation whether to publish the paper until I have seen a revised version.

Author Response
Please see word doc version for color distinction between your review and our responses.
ATI in the gas phase arguably is the best-investigated strong-field phenomenon in atomic physics. Of course, it provides the background to the current paper. The results for metals largely derive their interest by comparison with the atomic case. Yet, the paper does not contain a single reference to atomic ATI.
We’ve added several references to atomic ATI, both in text and citations.
In general, the list of references is completely inadequate. Just to begin with: Reference 1 is an unpublished 5-years' old Ph. D. thesis. Presumably, the important results of the thesis have been published in the mean time. Indeed, there is an extensive Topical Review (JPB 51, 172001 (2018)) with the thesis author on the list of authors. So why not cite this or other published papers by Wachter? Reference 2 is on HHG rather than ATI, it is focused on attosecond phenomena, which are irrelevant to the current paper and, moreover, it is hard to access, and so on ...
While we opted to keep the 5 year old thesis, as it provides a good background on the same topic and the Jellium potential (we only cite it when discussing the potential), we’ve also added several other citations.
There is also a lot of previous literature on ATI in metals. How does the present work fit in? What are the differences if any? Which problems unsolved as yet are now resolved?
We’ve added note of some other works on similar topics. What hasn’t been done yet is the extremely high enhanced fields that were unachievable with nanotips, and may now be possible with nanoblades. Such high field simulations have been performed with gases, but not metals.
The discussion of the model potential is unsatisfactory. The physical meaning of the various parameters should be discussed and not just their numerical values be given. So, what is the meaning of x_im, what are W, and r_s? Presumably, the potential plotted in Fig. 1 is V_jellium + V_s. If so, why not write it down, and what are the parameters underlying Fig. 1?
We’ve added the significance for all of the variables making up the potential. We also mention that the final potential is a superposition of V_jellium and V_s. Figure 1 is a rough schematic showing the blade system, there are no real parameters to be shown (except perhaps for the blade angle, which is now mentioned in 2.3).
The field (4) is like the incident field, except for a different space-dependent amplitude. What is the relation between the amplitudes of the incident field and the enhanced field at the tip?
It is stated that the enhancement factor is 12 for non-penentrating simulations and about 4 for FDTD-generated fields.
The authors compare their simulation results with "experiments", but do not disclose what these experiments are.
We’ve added citations to the paper (under review) whenever we mention those experiments.
Figure 7 exhibits "interesting hump structures", which the authors trace to "emissions earlier in the pulse" or to "perturbations from the semiclassical 3-step model approach". Actually, they are probably the same interference structures that are well known and have been much discussed in atomic ATI and can be traced to interference of different quantum orbits.
We’ve changed our explanation and added a citation to match on this subject.
What are the dashed red lines in Fig. 9 and what is the reason of the discrepancy between the atomic 10 U_p cutoff and the cutoffs of the metallic ATI spectra?
The explanation for the dashed red lines was accidentally left in Fig 10’s caption, and this is now fixed. We expect a slight discrepancy between 10 U_p and the actual spectrum because 10 U_p is based on a semiclassical approach, but we do not have an explanation for the large discrepancy for high wavelengths.
The total yields of Fig. 10 should be compared with the atomic case. There are many such calculations available in the published literature.
We’ve mentioned that atomic ATI achieves similar power laws.
The acknowledgment was written by the first author who expresses his appreciation of support by the other two. This is not the purpose of an acknowledgment.
We’ve adjusted the perspective of the acknowledgements to be from the authors collectively and not the first author.
We appreciate your thorough review of our manuscript.

Reviewer 2 Report
The article presented deals with a very current and interesting topic, namely the interaction of a material with a very intense electromagnetic field generated, for example, by a high-power and short-pulse laser beam. The electrons produced in these events can either be emitted or be rescattered.
The peculiarity of the work presented is the adoption of a blade nanostructure instead of a canonical tip shape. The disadvantage of the tip, from an operational point of view, is its fragile and low damage threshold. The blade, on the other hand, seems to be more robust and reliable thanks to a configuration more favorable to dissipation.
The authors present an interesting theoretical and simulation work to explore the potential of this new structure and, in fact, have achieved good results. Clearly, as stated by the authors, more work is needed to have a complete simulation and understanding of the phenomena. However, the results presented are encouraging and published in the north. I stress that the work is clearly written, and the authors already underline some of the weaknesses that require further investigation. Also interesting is the comparison between the VD and WO method for calculating the energy spectrum of the emitted electrons.
For my part, I have only a few comments:
on page 3, please explain the definition of Ef and W mention the Lumerical program 10. The caption of the figure, in the second period, refers to a red dashed line for the semi-classical cut that is not shown in the figure. Reference 4 refers to Optics Express, please correct
Author Response
Please see the attached word doc version for color coding to distinguish your review and our responses.
On page 3, please explain the definition of Ef and W…
We now explain E_f and W, among other variables.
…mention the Lumerical program.
It is unclear what you meant by this. The Lumerical simulation was run by the second author in-house, so there is no citation needed for it. There is enough information in the manuscript to reproduce this FDTD data.
The caption of the figure, in the second period, refers to a red dashed line for the semi-classical cut that is not shown in the figure.
That part of the caption was intended for Figure 9, and it now resides there.
Reference 4 refers to Optics Express, please correct
Good catch, we’ve adjusted the citation.
We thank you for your review.

Round 2
Reviewer 1 Report
The revised version of the paper is much improved. I recommend that it be published.